# A KAN-BASED LIGHTWEIGHT MODALITY FUSION METHOD FOR VIDEO-TEXT RETRIEVAL

## ABSTRACT

Different from the text-to-text retrieval tasks, video-text retrieval is significantly affected by the inherent modality difference between high-dimensional visual and textual data, which limits the model performance. Therefore, increasing works adopted the modality fusion techniques to effectively improve the model accuracy, while the attention mechanism from the Transformer is widely adopted to improve the accuracy. However, the high quadratic computational complexity from the attention mechanism generates the prohibitive memory cost, which is the obstacle to the effective training on the machine and inference stage in the real world. Therefore, to tackle the challenge, this paper proposes KFusion to fuse text and video frames with lower computational cost, achieved by employing the Kolmogorov-Arnold-Network-based Bridge module and Text-Frame Mamba. Bridge captures the cross-modal feature via the learnable spline-based activation functions. It calculates the weights for the text and video to facilitate the video-text fusion, but the unimportant information from the text and video hampers the fusion effect. Therefore, Text-Frame Mamba contains separate Mamba backbones, which is proposed to remove the noise from the important text and frame embeddings through the state space models. The weight vectors calculated by the Bridge multiply the filtered information processed by the Mamba backbones. The performances on the MSRVTT, MSVD and Didemo benchmark datasets demonstrate the state-of-the-art performance of KFusion in terms of the accuracy and efficiency.

## 1 INTRODUCTION

Video-text retrieval (VTR) is a critical task in the multi-modal understanding, divided into video-to-text and text-to-video retrieval. Thanks to the surge of the online video, VTR has been attracting a significant attention. The models in VTR retrieves the relevant text descriptions for videos given the cross-modal similarity, and vice versa.

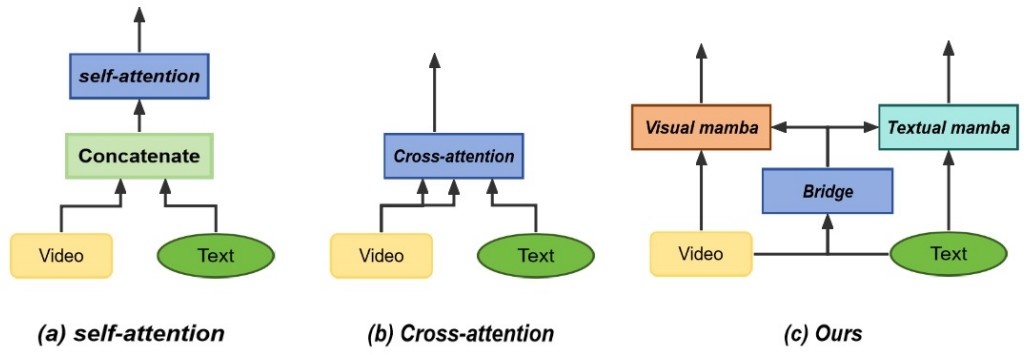

Figure 1: Architecture and performance comparisons of modality fusion methods, including self-attention, cross-attention, and our proposed KFusion.

Current VTR research mainly adopted dual-stream and single-stream model. The single-stream modelsDzabraev et al. (2021) achieved higher accuracy by jointly processing the visual and textual inputs, but the issue of the computation efficiency hampered their scalability. Meanwhile, the dual-stream architectures process the visual and textual modalities independently. Those works Zhu & Yang (2020) outperformed the single-stream model in the VTR in terms of the efficiency, but the insufficient interaction between the text and video limit the accuracy compared to the single-stream models. However, since the emergence of the CLIP Radford et al. (2021) pre-trained on 400-million image-text pairs has changed the paradigm due to its zero-shot performance. Thereby, increasing research began to fine-tune on the CLIP, and they significantly outperformed non-CLIP-based approaches Dzabraev et al. (2021); Li et al. (2022).

However, the zero-shot ability of the CLIP is still constrained by the inherently-huge modality gap between visual and textual representations that degrades the accuracy of the alignment. To bridge the gap between the text and video, as shown in Figure 1, many researchers adopted attention mechanism from Transformer Vaswani et al. (2017) to fuse the visual information and text and achieved higher accuracy, mainly divided into the cross-attention and self-attention. Those works that fuse the textual and visual embedding Chen et al. (2023); Wu et al. (2023) took the concatenated the textual and visual embedding as the input to the self-attention. Those works Wang et al. (2024); Jin et al. (2023b;a); Gorti et al. (2022) that applied cross-attention used the text as the query vector, and the visual embedding as the key and value vectors to make the visual embedding fused with the textual information.

However, in the stage of the real-world inference and the fine-tuning on the datasets, the interaction between the text and video leads to a high memory usage. The attention mechanism from the Transformer inevitably suffers from the quadratic space complexity, which limits the further feasible exploration on the use of Transformer regarding the modality fusion.

Therefore, this paper proposes the KFusion adopting Mamba and Vim as the backbone to fuse the modality, consisting of a Bridge based on the Kolmogorov-Arnold Network (KAN)Liu et al. (2024) and the Text-Frame Mamba.

Compared to the attention mechanism, KAN is a lightweight module that generate lower memory consumption. KAN calculates the weight vectors for the text and video, achieved by processing the joint text and video. The weight vectors for the text and video is multiplied with the filtered text and video features. However, KAN is insensitive to the positional information that captures the relative position of the text and video frames. Therefore, the positional information of the text and video frames will be respectively injected given the relative positions into the text and video embedding before the concatenation.

However, the modality weights are affected by the noisy features from the text and video, hampering the effect of the modality fusion, which can also be shown in table 4. Therefore, to optimize the effect of fusion, it is necessary to filter the less contributory information.

As the aforementioned disadvantage regarding the computational efficiency, the filtration of the irrelevant information is necessary. Recently, MambaGu & Dao (2023) achieved superior accuracy over Transformer with linear computational space complexity. Mamba was first used in 1-D textual and DNA information modeling. Subsequently, increasing works Zhu et al. (2024); Li et al. (2024); Tang et al. (2024); Liu et al. (2025) also extended Mamba to the 2-D computer vision and even multi-modal downstream tasks.

Bridge replaces the residual connections from the original Mamba and Vim branch. Text-Frame Mamba applies the textual Mamba based on Mamba Gu & Dao (2023) for 1-D text and visual Mamba based on Vim Zhu et al. (2024) for 2-D sampled video frames as the backbone to optimize the textual and visual features from via the state space models (SSM), respectively. As shown in the Table 4, applying the separate Mamba frameworks shows the higher accuracy over the way that using two identical Mamba framework. The experimental result demonstrated the advantage of using modality-specific Mamba backbones in the multi-modal domain.

The main contributions can be summarized as follows:

(1) The Bridge integrates the relative positional information to produce the weights to fuse the video and text, along with the filtered features produced from the Text-Frame Mamba.

(2) This Text-Frame is proposed to remove the noisy text and video features through the state space models. KFusion includes the modality-specific Mamba backbones for the effectiveness to discard the insignificant video and text features.

(3) Evaluated on extensive benchmark datasets, KFusion demonstrates the effectiveness to fuse the modality with the consideration of the efficiency.

## 2 RELATED WORKS

### 2.1 VIDEO-TEXT RETRIEVAL

Followed the emergence of the CLIP Radford et al. (2021), CLIP4Clip Luo et al. (2022) endeavored to fine-tune the CLIP on the video-text benchmark and demonstrates the better performance compared to the previous non-CLIP-based works because of the outstanding zero-shot ability of the CLIP. Subsequent works Liu et al. (2022); Ma et al. (2022); Wang et al. (2022) replaced the coarse-grained alignment from the CLIP4Clip with the fine-grained alignment between the text and video, which improved the accuracy of the model. XCLIP Ma et al. (2022) proposed the attention over similarity matrix to calculate the similarity between the global and local representation. DRL Wang et al. (2022) proposed the weighted-token-wise maximum based on the mean max strategy from the Colbert Khattab & Zaharia (2020). Besides, The video representation is aggregated by the video frame embeddings. Nonetheless, not every frame contributes equally. Thus, works Wang et al. (2022); Buch et al. (2022); Liu et al. (2023); Fang et al. (2022) advanced the temporal modeling to weight more contributory frames. However, the aforementioned works did not reduce the modality gap, which inherently impairs the performance in every multi-modal tasks. Since the modality fusion is an effective approach to mitigate the modality gap, works Wang et al. (2024); Jin et al. (2023a); Gorti et al. (2022) adopted the cross-attention mechanism based on the attention mechanism to integrate the textual information into the video frames. TABLE Chen et al. (2023) concatenated the text information with the video frames then fed to the self-attention. Those works outperform the CLIP-based works that do not fuse the modality. However, the attention mechanism from the transformer inevitably leads to the high memory consumption, which will compromise the computational efficiency in the training and inference stage. Therefore, this paper adopts a Kolmogorov-Arnold-Network-based Bridge combined with Mamba to fuse the modality.

### 2.2 MAMBA

Mamba was first adopted in the language tasks, outperforming the Transformer in terms of the accuracy with the less complexity and parameters. The state space model (SSM) from the Mamba capture the important information from a long-range sequence. Subsequently, researchers Zhu et al. (2024); Liu et al. (2025) exploited the Mamba backbone and proposed the bidirectional mamba, extending the Mamba to the 2-D computer vision task. VideoMamba Li et al. (2024) followed the similar theories Radford et al. (2021); Dosovitskiy et al. (2020); Zhu et al. (2024), which extended the Mamba to the 3-D video understanding. Muse Tang et al. (2024) is integrated with a residual network to model the joint resolution to produce the visual features, using the Vim Zhu et al. (2024). VMamba Liu et al. (2025) proposed a four-way scanning mechanism tailored for spatial domain traversal, enabling each image patch to gain contextual knowledge exclusively via a compressed hidden state computed along the corresponding scanning path. MambaSODZhan et al. (2025) fused the RGB and Depth through replacing the linear layers used for the residual connection with the SSM branch, but they reside in the similar embedding space, which is not feasible to adopt the theory in the video-text or image-text works. Thereby, this work proposes Mamba and Vim as the backbones to filter the unnecessary information from the text and video, respectively, which significantly increased the accuracy compared to the widely-used attention mechanism.

## 3 METHODOLOGY

As shown in the Figure 2, this section elaborates the architecture of the KFusion, including the video and text encoder in Section 3.1, the Bridge to fuse the modality in Section 3.2, the Text-Frame Mamba in the Section 3.3, and the training objective in Section 3.4.

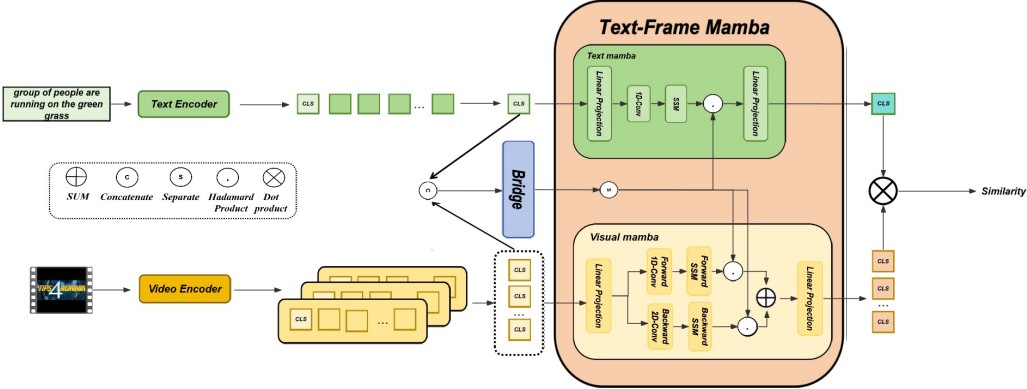

Figure 2: The model architecture of the KFusion, which includes the video encoder, text enocder, Bridge and Text-Frame Mamba.

## 3.1 VIDEO AND TEXT ENCODER

KFusion utilizes a lower-case byte pair encoding (BPE) tokenizer with a vocabulary of 49,152 to tokenize the passage $T = \{t_1, t_2, \ldots, t_{N_t}\}$, which is then processed using the Transformer model from the CLIP, where $N_t$ denotes the word count from a passage. [CLS] is the global representation.

Meanwhile, the Vision Transformer (ViT) from CLIP is employed to encode sampled video frames $F$ from $V = \{F_1, F_2, \ldots, F_n\}$, where $n_f$ indicates the total number of frames. Both ViT-B/32 and ViT-B/16 are employed as our Video Encoders to encode the raw video. Here, $B$, $F$ and $E$ represent the batch size, the number of sampled frames per video, and the embedding size, respectively.

## 3.2 BRIDGE

To fuse the modalities, it is very important to apply a Bridge. As shown in Figure 2, to fuse the modalities, KFusion replaces the linear layer on the residual connection branches from the textual and visual Mamba with KAN, which outperformed MLP because of the interpretability and the mitigation on the catastrophic forgetting. Specifically, as shown in Figure 5, KAN outperforms the other models in terms of accuracy. As shown in Figure 3, before concatenating to feed to the Bridge, text and video are summed with their positional embeddings $T_E$ and $F_{E_i}$, formulated as:

$$T_E = \sin\left(\frac{\text{pos}}{10000}\right) \tag{1}$$

$$F_{E_{2i+1}} = \sin\left(\frac{\text{pos}}{10000^{2i/d}}\right) \tag{2}$$

$$F_{E_{2i}} = \cos\left(\frac{\text{pos}}{10000^{2i/d}}\right) \tag{3}$$

Then, $F_{E_i}$ and $T_E$ are summed with the original video frame and text embeddings for further concatenation, which can be formulated as:

$$F_{i_y} = F_{E_i} + F_i \quad i = 1, \ldots, N \tag{4}$$

$$T_y = T_E + T \tag{5}$$

$$M = \text{concat}(T, F_1, F_2, \ldots, F_N) \tag{6}$$

Where concat, $T_y$, $F_{iy}$, and $M$ denote the concatenation operation, the textual and video frame embeddings that incorporate their positional information, and the concatenated modality information, respectively. Subsequently, $M$ will be fed to the Bridge to obtain the weight vector, which is implemented by separating the output.

KAN is applied between two layers for normalization. Different from the MLP models, KAN primarily updates the parameters from the activation functions rather than the weights, formulated are:

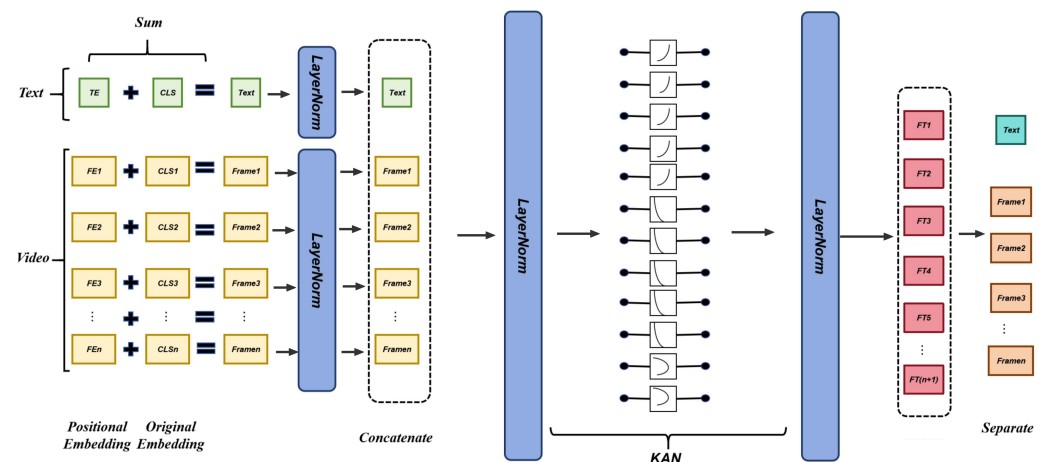

Figure 3: The Bridge is based on KAN, interleaved with several layer normalization layers. It fuses the modality between the video and text.

$$M_y = \Phi \mathbf{o} M = \left[ \sum_{i=1}^{d_{\text{in}}} \phi_{i,1}(M_i) \cdots \sum_{i=1}^{d_{\text{in}}} \phi_{i,d_{\text{out}}}(M_i) \right] \tag{7}$$

where $\Phi$ is produced by the $\Phi_{\text{in}}$ and $\phi_{\text{out}}$, which can be defined as:

$$\phi = \begin{pmatrix} \Phi_1 \\ \Phi_2 \\ \vdots \\ \Phi_{2n+1} \end{pmatrix} \begin{pmatrix} \phi_{1,1} & \phi_{1,2} & \cdots & \phi_{1,2n+1} \\ \phi_{2,1} & \phi_{2,2} & \cdots & \phi_{2,2n+1} \\ \vdots & \vdots & \ddots & \vdots \\ \phi_{n,1} & \phi_{n,2} & \cdots & \phi_{n,2n+1} \end{pmatrix} \tag{8}$$

where $\phi$ is the combination of the SiLU activation and B-Spline function, which can be formulated via the linear combination as:

$$\phi(M) = w_b \frac{M}{1 + e^{-M}} + w_s \sum_i c_i B_i(x) \tag{9}$$

Here, $B_i(x)$ is a B-spline function. $w_b$ and $w_s$ denote the weight parameters, and $c_i$ is a control coefficient to shape the B-spline that represents any univariate function on a finite domain.

After passing the KAN, the KFusion separates the concatenated text and video to obtain the weights for the text and video, which are $w_t$ and $w_v$, respectively.

### 3.3 TEXT-FRAME MAMBA

Since Mamba demonstrates the better performance than transformer in terms of the memory and accuracy, as shown in Figure 2, this paper proposes Text-Frame Mamba that utilizes two modality-specific Mamba backbones to fuse the video and text.

However, different from the existing works that applied a single Mamba backbone, the Text-Frame Mamba from the KFusion leveraged the Vim Zhu et al. (2024) and Mamba Gu & Dao (2023) to respectively prioritize the more contributory information from the 1-D text and 2-D frame feature via the SSM to solve the issue of modality adaptability.

It maps the input sequence $x_t$ and $x_v$ to the output sequence via the hidden state $h_t(x)$ and $h_v(x)$, evolution parameters $A_t \in \mathbb{R}^{N \times N}$ and $A_v \in \mathbb{R}^{N \times N}$, projection parameters $B_t \in \mathbb{R}^{N \times 1}$, $B_v \in$

$\mathbb{R}^{N \times 1}$, $C_t \in \mathbb{R}^{1 \times N}$ and $C_v \in \mathbb{R}^{1 \times N}$, which can be formulated as:

$$h'_t(T) = A_t h_t(T) * B_t x_t(T) \tag{10}$$

$$h'_v(V) = A_v h_v(V) * B_v x_v(V) \tag{11}$$

$$T_y = C_t h(T) \tag{12}$$

$$V_y = C_v h(V) \tag{13}$$

However, the parameters are continuous. Only when the parameters $A_t$, $A_v$, $B_t$ and $B_v$ are discretized through the timescale $\Delta$ to produce $\overline{A_t}$, $\overline{A_v}$, $\overline{B_t}$ and $\overline{B_v}$, respectively, the Mamba can be used in the training and inference, which is calculated as:

$$\overline{A_t} = \exp(\Delta A_t) \tag{14}$$

$$\overline{B_t} = (\Delta A_t)^{-1}(\exp(\Delta A_t) - I)\Delta B_t \tag{15}$$

$$\overline{A_v} = \exp(\Delta A_v) \tag{16}$$

$$\overline{B_v} = (\Delta A_v)^{-1}(\exp(\Delta A_v) - I)\Delta B_v \tag{17}$$

As shown in Figure 2, since the video frame is 2-D features, the Vim Zhu et al. (2024) that proposed the backward direction based on the Mamba is leveraged for frames. Despite this modification, the state space models from the visual Mamba and Mamba can select the critical textual and visual details effectively to produce $T_y$ and $V_y$, respectively, which can be formulated as:

$$T_y = T * \overline{K_t} \tag{18}$$

$$V_y = V * \overline{K_v} \tag{19}$$

Where $\overline{K_t}$ and $\overline{K_v}$ are the coefficient convolutional details in the textual and visual mamba, respectively, which can be formulated as:

$$\overline{K_t} = (C_t\overline{B_t}, C_t\overline{A_t B_t}, \dots, C_t\overline{A_t^{L-1}B_t}, \dots, C_t\overline{A_t^{N_t-1}B_t}) \tag{20}$$

$$\overline{K_v} = (C_v\overline{B_v}, C_v\overline{A_v B_v}, \dots, C_v\overline{A_v^{L-1}B_v}, \dots, C_v\overline{A_v^{n_f-1}B_v}) \tag{21}$$

Then, the refined text feature $T_y$ and video feature $V_y$ produced from Text-Frame Mamba, which can be formulated as:

$$T' = T_y \circ w_t \tag{22}$$

$$V' = V_y \circ w_t \tag{23}$$

Where $V'$ and $T'$ are the fused video frame and text features, respectively. Then the fused video frame embeddings will be aggregated to the overall video representation $\overline{V'}$ via mean pooling to produce the similarity matrix $S(V, T)$, which can be formulated as:

$$\overline{V'} = \frac{1}{n}\sum_{i=1}^{n} V'_i \tag{24}$$

$$S(V, T) = \frac{\overline{V'} \cdot T'}{\|\overline{V'}\| \cdot \|T'\|} \tag{25}$$

## 3.4 TRAINING OBJECTIVE

In this study, the InfoNCE loss is applied to maximize the similarity values on the diagonal, while reducing the similarity values of unrelated pairs, which can be formulated as:

$$L_{v2t} = -\frac{1}{B} \sum_{i=1}^{B} \log \left( \frac{\exp(\tau \cdot s(v_i, t_i))}{\sum_{j=1}^{B} \exp(s(v_i, t_j))} \right) \tag{26}$$

$$L_{t2v} = -\frac{1}{B} \sum_{i=1}^{B} \log \left( \frac{\exp(\tau \cdot s(v_i, t_i))}{\sum_{j=1}^{B} \exp(s(v_j, t_i))} \right) \tag{27}$$

Where $L_{v2t}$ and $L_{t2v}$ denote the InfoNCE loss in the video-to-text and text-to-video directions, respectively. Then, the overall objective InfoNCE loss is computed by averaging $L_{v2t}$ and $L_{t2v}$:

$$L_{InfoNCE} = \frac{1}{2} \left( L_{v2t} + L_{t2v} \right) \tag{28}$$

Here, $L_{InfoNCE}$ targets the optimization of cross-modal similarities, with $B$ representing the batch size and $\tau$ the temperature hyper-parameter.

## 4 EXPERIMENT

### 4.1 BENCHMARK AND EVALUATION METRICS

In this paper, to evaluate the effectiveness of the KFusion, experiments were conducted on MSRVTT Xu et al. (2016), MSVD Chen & Dolan (2011), and DiDeMoAnne Hendricks et al. (2017)datasets.

The MSR-VTT dataset includes 10,000 videos, with a duration ranging from 10 to 32 seconds and 20 captions per video. The experiments used a training set of 9,000 videos and a test set of 1,000 text-video pairs. The MSVD dataset contains 1,970 videos, with train, validation, and test splits comprising 1,200, 100, and 670 videos, respectively, and about 40 sentences per video in English. DiDeMo contains 10,000 videos annotated with 40 sentences each, concatenating all text descriptions into a single passage per video.

Standard retrieval metrics are recall at rank K (R@K), median rank (MdR), and mean rank (MnR) , which were used to evaluate the methodology. R@K evaluates model performance by the matched samples among the top K results. The paper used K=1,5, and 10 as retrieval criteria. MdR and MnR measure the median and mean positions of the matched results, respectively, where lower values indicate better performance.

### 4.2 EXPERIMENT DETAILS

Experiments were conducted on 8 NVIDIA GeForce RTX 4090 GPUs in 35 hours by the PyTorch library. Following practices from previous CLIP-based works, the Transformer and Vision Transformer from CLIP are initialized as the text encoder and video encoder, respectively. The Adam optimizer is used with weight decay regularization, and the learning rate is decayed following a cosine schedule. The initial learning rates are set as 1e-7 for the text and video encoder, and 1e-3 for other modules. The values of other parameters vary given the specific properties of a benchmark. For the maximum sentence length, we set 32 for MSR-VTT and MSVD, and 64 for ActivitityNet, and Didemo. For the maximum total frames in a video, we assign 12 for MSR-VTT, MSVD, and 64 for ActivityNet, and Didemo. Batch sizes are set to 256 for MSR-VTT, MSVD, and 512 for ActivityNet, and Didemo. Each benchmark is run for 5 training epochs.

### 4.3 COMPARISONS TO STATE-OF-THE-ART METHODS

Table 1 presents the performance of the KFusion on the MSRVTT dataset. In the text-to-video retrieval, when using ViT-B /32 as the backbone, the KFusion surpassed the TABLE and TS2-Net by 0.4% and 0.5% on the metric R@1, respectively. Moreover, it surpassed the X-Pool by 0.6% on

it. Meanwhile, when using ViT-B/16 as the backbone, on the metric R@1, the KFusion achieved 51.8% on the text-to-video (T2V) retrieval, exceeding STAN by 1.8%, and 51.1% on the video-to-text (V2T) retrieval and overtook the UcoFiA by 2.0%.

Table 1: Performance comparison on MSRVTT dataset.

| Model | Text-to-Video Retrieval | | | | | Video-to-Text Retrieval | | | | |
|---|---|---|---|---|---|---|---|---|---|---|
| | R@1 | R@5 | R@10 | MdR | MnR | R@1 | R@5 | R@10 | MdR | MnR |
| *CLIP-based models (ViT-B/32)* | | | | | | | | | | |
| CLIP4CLIP Luo et al. (2022) | 44.5 | 71.4 | 81.6 | 2.0 | 15.3 | 42.7 | 70.9 | 80.6 | 2.0 | 11.6 |
| X-CLIPMa et al. (2022) | 46.1 | 73.0 | 83.1 | 2.0 | 13.2 | 46.8 | 73.3 | 84.0 | 2.0 | 9.1 |
| XPoolGorti et al. (2022) | 46.9 | 72.8 | 82.2 | 2.0 | 14.3 | 44.3 | 73.3 | 84.0 | 2.0 | 9.0 |
| TS2-Net Liu et al. (2022) | 47.0 | 74.5 | 83.8 | 2.0 | 13.0 | 45.3 | 74.1 | 83.7 | 2.0 | 9.2 |
| TABLE Chen et al. (2023) | 47.1 | 74.3 | 82.9 | 2.0 | 13.4 | 47.2 | 74.2 | 84.2 | 2.0 | 11.0 |
| DRL Wang et al. (2022) | 47.4 | 74.6 | 83.8 | 2.0 | 12.8 | 45.3 | 73.9 | 83.3 | 2.0 | - |
| DiCoSAJin et al. (2023a) | **47.5** | 74.7 | 83.8 | 2.0 | 13.2 | 46.7 | 75.2 | 84.3 | 2.0 | **8.9** |
| KFusion (ViT-B/32) | **47.5** | **75.6** | **84.6** | **2.0** | **12.2** | 47.1 | **76.1** | **84.7** | **2.0** | 9.3 |
| *CLIP-based models (ViT-B/16)* | | | | | | | | | | |
| CLIP4CLIP Luo et al. (2022) | 46.4 | 72.1 | 82.0 | 2.0 | 13.3 | 45.4 | 73.4 | 82.4 | 2.0 | 10.8 |
| CenterCLIPZhao et al. (2022) | 48.4 | 73.8 | 82.0 | 2.0 | 13.8 | 47.7 | 75.0 | 83.3 | 2.0 | 10.2 |
| X-CLIPMa et al. (2022) | 49.3 | 75.8 | 84.8 | 2.0 | 12.2 | 48.9 | 76.8 | 84.5 | 2.0 | 8.1 |
| TS2-Net Liu et al. (2022) | 49.4 | 75.6 | 83.8 | 2.0 | 13.5 | 46.6 | 75.9 | 84.9 | 2.0 | **8.9** |
| DRL Wang et al. (2022) | 50.2 | 76.5 | 84.7 | **1.0** | 12.4 | 48.9 | 76.3 | 85.4 | 2.0 | - |
| KFusion (ViT-B/16) | **51.8** | **78.3** | **85.8** | 2.0 | **10.9** | **51.1** | 76.8 | 86.3 | 2.0 | 8.6 |

Table 2: Retrieval performance comparisons on MSVD dataset.

| Model | Text-to-Video Retrieval | | | Video-to-Text Retrieval | | |
|---|---|---|---|---|---|---|
| | R@1 | R@5 | R@10 | R@1 | R@5 | R@10 |
| CLIP4CLIPLuo et al. (2022) | 46.2 | 76.1 | 84.6 | 56.6 | 79.7 | 84.3 |
| DiffusionRetJin et al. (2023b) | 46.6 | 75.9 | 84.1 | 61.9 | 88.3 | 92.9 |
| X-CLIPMa et al. (2022) | 47.1 | 77.8 | - | 60.9 | 87.8 | - |
| X-PoolGorti et al. (2022) | 47.2 | 77.4 | 86.0 | 66.4 | **90.0** | 94.2 |
| CenterCLIPZhao et al. (2022) | 47.3 | 76.9 | 86.0 | 63.5 | 86.4 | 92.6 |
| TABLE Chen et al. (2023) | 47.3 | **77.4** | 85.5 | **68.9** | 93.1 | 97.1 |
| KFusion (Ours) | **48.0** | 76.4 | **87.2** | 68.1 | 89.6 | **97.3** |

Table 3: Retrieval performance comparisons on DiDeMo dataset.

| Model | Text-to-Video Retrieval | | | Video-to-Text Retrieval | | |
|---|---|---|---|---|---|---|
| | R@1 | R@5 | R@10 | R@1 | R@5 | R@10 |
| CLIP4CLIPLuo et al. (2022) | 43.4 | 70.2 | 80.6 | 43.4 | 69.9 | 80.2 |
| X-CLIPMa et al. (2022) | 45.2 | 74.0 | - | 43.1 | 72.2 | - |
| DiCoSAJin et al. (2023a) | 45.7 | 74.6 | **83.5** | - | - | - |
| VopHuang et al. (2023) | 46.4 | 71.9 | 81.5 | 44.4 | 71.8 | 81.8 |
| UcoFiAWang et al. (2023) | 46.5 | 74.8 | 84.4 | 46.0 | 71.9 | 81.5 |
| KFusion (Ours) | **46.6** | **75.4** | 82.1 | **46.1** | **74.3** | **88.8** |

In table 2, on the MSVD datatset, KFusion surpassed X-Pool by 0.8% on the metric R@1 and 0.3% in the T2V and V2T retrieval, respectively. In table 3, on the DideMo dataset, KFusion achieves 46.6% and 46.1% on the metric R@1 in the T2V and V2T retrieval, respectively.

Meanwhile, it shows the memory usage and accuracy of KFusion on the MSRVTT datatset compared to X-Pool that applies the cross-attention mechanism, and CLIP4Clip that does not apply any multimodal fusion approach. During the training and inference stage, even though the fusion technique from the KFusion leads to 55.2 GB and 56.8 GB higher memory consumption, respectively, the KFusion surpasses the CLIP4CLIP in terms of accuracy. Moreover, the KFusion achieves 0.6% higher accuracy with 23.2 GB lower memory usage in the inference stage. As depicted in the table, the KFusion surpasses the X-Pool with 0.8 % higher accuracy with 28.8 GB lower memory consumption in the inference stage.

Compared to MSRVTT, the datasets from DiDeMo and MSVD are much smaller, which leads to overfitting and causes Mamba to relatively lag behind the state-of-the-art models. However, the proposed KFusion demonstrates the advantages over the existing works regarding the accuracy and memory use during the inference and training stages.

## 4.4 ABLATION STUDIES

*Modules* As shown in Table 4, this ablation study also analyzes the modules from the KFusion in contrast to the CLIP4Clip. Only applying the Bridge to fuse the modality is inferior to the CLIP4Clip. When applying different Mamba-based backbones to process the video frames and text separately without considering the modality adaptability, its performance was slightly better than applying a Mamba for text or Vim for video frames only.

Table 4: Performance for different modules. POS stands for the positional embedding given the modality, where $2 \times$ stands for applying Mamba or Vim to text and video.

| Model | Text-to-Video | | |
| --- | --- | --- | --- |
| | **R@1** | **R@5** | **R@10** |
| X-Pool | 46.9 | 72.8 | 82.2 |
| CLIP4Clip | 43.1 | 70.4 | 80.8 |
| + Bridge | 38.4 | 57.1 | 70.4 |
|   + POS | 40.5 | 63.2 | 74.9 |
|     + Mamba | 42.8 | 69.1 | 82.5 |
|     + Vim | 42.6 | 69.8 | 83.1 |
|     + $2 \times Mamba$ | 44.2 | 72.2 | 82.5 |
|     + $2 \times Vim$ | 42.6 | 68.2 | 78.7 |
|     + Text-Frame Mamba | **47.5** | **75.6** | **84.6** |

*Bridge* Apart from the KAN, this section analyzes the feasibility of self-attention and MLP to fuse the modality for the ridgebackbone. As shown in Table 5, improvement on the accuracy shows the necessities to fuse the video and text. Because of the model complexity that leads to overfitting, when applying the self-attention mechanism from the Transformer, it falls behind the MLP. KAN surpasses MLP by 0.7% on the R@1 at the text-to-video retrieval.

Table 5: Performance comparison between the possible models that fuse the modality, where NP stands for not replacing linear layer for the residual connection.

| Module | Text-to-Video | | |
| --- | --- | --- | --- |
| | **R@1** | **R@5** | **R@10** |
| MLP | 46.8 | 75.8 | 84.4 |
| self attention | 46.4 | 75.4 | 83.9 |
| KAN | 47.5 | 75.6 | 84.6 |

*Layer numbers of the Text-Frame Mamba* As shown in Table 6, even though applying the Text-Frame Mamba outperforms the Transformer with regard to the accuracy and space efficiency, more layers can produce the higher accuracy. However, with the increase of the Text-Frame layers, the model accuracy started to decrease, which can be attributed to the introduction of the massive parameters that complicates the model, leading to the overfitting.

Table 6: Performance comparison about the layer numbers of the Text-Frame Mamba

| Layer number | Text-to-Video | | |
| --- | --- | --- | --- |
| | **R@1** | **R@5** | **R@10** |
| 1 | 47.2 | **76.1** | 83.8 |
| 2 | **47.5** | 75.6 | **84.6** |
| 3 | 46.9 | 73.3 | 82.3 |
| 4 | 45.4 | 68.9 | 80.1 |

## 5 CONCLUSION

KFusion significantly fuses the modality in video-text retrieval to achieve higher accuracy with the lower computational consumption, which is accomplished by the KAN that fuses the text and video by producing the weights, and the Text-Frame Mamba leveraging the modality-specific Mamba backbones to discard the information for the better fusion effect. The experiments on multiple benchmark datasets demonstrate that KFusion outperforms existing methods in terms of accuracy and computational efficiency.

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

## A  APPENDIX

### A.1  QUALITATIVE ANALYSIS

In this section, the videos and texts are from the MSRVTT dataset, using the X-Pool for comparison. As shown in the Figure 4, in the text-to-video retrieval, in the example (a), KFusion retrieves the video that contains the "blue dree" and "gloves", but no glove is shown in the video retrieved by the X-Pool. In the example (c), KFusion retrieved the video that contains the key word "satelite", while X-Pool retrieved a video containing a man without the "satelite".

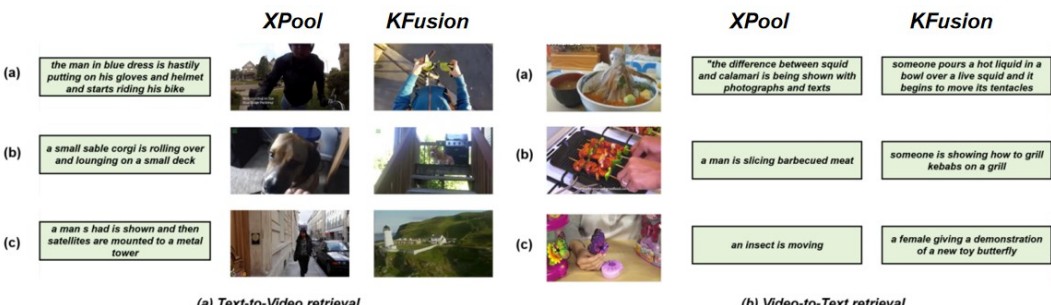

Figure 4: The qualitative analysis via comparing KFusion with X-pool for the text-video retrieval, divided into the text-to-video retrieval and video-to-text retrieval.

In the V2T retrieval, in the example (b), the text retrieved by X-pool contains "slicing" that does not depicted in the video file, but the text retrieved by the KFusion contains "grill" and "kebabs". Meanwhile, in the example (c), the word "insect" and "moving" from the text retrieved by the X-Pool were not corresponding to the video, but the "toy butterfly" from the text retrieved by the KFusion suits more.

