# OpenReview forum: "A KAN-based lightweight modality fusion method for video-text retrieval"
_ICLR.cc/2026/Conference — ICLR 2026 Conference Withdrawn Submission_

### Official Review · Reviewer_U2Ks · 2025-10-27

**Soundness:** 1
**Presentation:** 1
**Contribution:** 1
**Rating:** 2
**Confidence:** 5

**Summary:**

This paper presents KFusion, a lightweight modality fusion method for video-text retrieval (VTR). The challenge in VTR arises from the inherent differences between visual and textual data, which affects model performance. The paper proposes a solution that combines Kolmogorov-Arnold-Network-based Bridge (KAN) and Text-Frame Mamba to reduce computational cost while improving fusion accuracy. KFusion addresses the high memory consumption of Transformer-based attention mechanisms and achieves efficient modality fusion by using KAN for cross-modal weight calculation and Mamba backbones for filtering noisy features. The approach is validated through experiments on popular datasets, showing superior performance compared to existing methods.

**Strengths:**

**Comprehensive Evaluation**: The paper includes extensive experiments across different datasets (MSRVTT, MSVD, DiDeMo), ablation studies, and comparisons to state-of-the-art methods, providing validation for the proposed method.

**Weaknesses:**

**Complexity in Model Setup**: The combination of KAN and Mamba with positional embeddings introduces a relatively complex model architecture, which might pose challenges in terms of scalability.

**Trivial Method**: This paper simply integrated Mamba and KAN to build a model, without much insights.

**Poor Performance**: The performance is poor and do not compare recent methods such as Cap4Video [1].

[1] Wu, Wenhao, Haipeng Luo, Bo Fang, Jingdong Wang, and Wanli Ouyang. "Cap4video: What can auxiliary captions do for text-video retrieval?." In Proceedings of the IEEE/CVF conference on computer vision and pattern recognition, pp. 10704-10713. 2023.

**Questions:**

N/A

---

### Official Review · Reviewer_KKkF · 2025-10-28

**Soundness:** 3
**Presentation:** 2
**Contribution:** 3
**Rating:** 4
**Confidence:** 4

**Summary:**

While the method proposed by the authors addresses some issues existing in video-text cross-modal retrieval models, the paper still has several aspects that need improvement. First, all the problems claimed to be solved in the paper must be supported by specific experiments or data to verify the effectiveness of the solutions. Second, the currently selected baseline methods lack timeliness; it is recommended to supplement the latest research from 2024 and 2025 as baselines to better demonstrate the cutting-edge nature of this study. In addition, the authors should pay attention to the layout aesthetics of the tables, optimize the image quality to ensure that the images remain clear and free from distortion when enlarged, and thereby enhance the visual presentation effect of the paper.

**Strengths:**

In terms of originality, the paper innovatively integrates KAN (Kolmogorov-Arnold Network) with modality-specific Mamba to construct the KFusion framework. It supplements positional information via the KAN-Based Bridge and adapts to text and video features through the Text-Frame Mamba, addressing the accuracy-efficiency trade-off in the field of Video-Text Retrieval (VTR). However, the paper has some issues that require further refinement by the authors.

**Weaknesses:**

1. The paper mentions that "KAN is a lightweight module that reduces memory consumption", yet this advantage is not demonstrated in the experimental section. It is recommended to supplement comparative data on model encoding time or parameter scale to intuitively showcase the "low memory consumption" feature.

2. It is suggested to reorganize the structure of the Introduction section. The current paragraph division is relatively scattered; adjusting the logical flow of paragraphs can enhance the connection between content, thereby improving overall coherence and organization.

3. In Section 3.1, should $F_{n}$ in the set $V=\left \{ F_{1},F_{2},...,F_{n} \right \}$ be revised to $F_{n_{f}}$? Additionally, the layout of Tables 4, 5, and 6 can be further optimized to improve aesthetics and readability.

4. Currently, all the baseline methods used for comparison are studies before 2023. As a conference paper, it is recommended to supplement comparisons with the latest methods from 2025 to more comprehensively demonstrate the timeliness and advancement of the research.

**Questions:**

Please refer to the Weaknesses.

---

### Official Review · Reviewer_9krg · 2025-10-30

**Soundness:** 2
**Presentation:** 1
**Contribution:** 2
**Rating:** 4
**Confidence:** 5

**Summary:**

This paper proposes KFusion, a lightweight video–text fusion framework that combines a Kolmogorov–Arnold-Network (KAN)–based Bridge (to produce adaptive per-modality weights) with a Text-Frame Mamba module (two modality-specific SSM backbones for text and frames). The Bridge outputs are applied to reweight text and video features before a CLIP-style InfoNCE training objective is used. Experiments are reported on MSR-VTT, MSVD, and DiDeMo with additional ablations on module choices and layer depth.

**Strengths:**

1. Clear motivation to reduce attention’s memory cost for cross-modal alignment and to handle modality-specific signal filtering with SSMs.

2. Modular design: separating Bridge (weighting) and Text-Frame Mamba (filtering) is conceptually neat and aligns with CLIP-style retrieval training.

3. Useful ablations: Tables 4–6 compare Bridge variants (MLP, self-attention, KAN) and Mamba layer depth; KAN slightly outperforms MLP; 2-layer Text-Frame Mamba is best among tested depths.

**Weaknesses:**

1. The writing quality of this paper is poor, making it difficult to read. The Introduction section contains too many paragraphs, and the frequent use of transition words such as "however" disrupts the logical flow of the narrative.

Typos:
In page 4, inside the description of “Figure 2”, the “text enocder” should be changed to “text encoder”.
In page 4, line 194, where is Figure 5?
In page 7, line 370~372, “ActivitytyNet” should be “ActivtyNet” if you are going to include ActivityNet
In page 8, line 418 and 421, two “datatset” typos.
In page 9, line 453, typo for “rdigebackbone”.
In page 7 equation (26, 27), $\tau$ is missed in the denominator.

2. The font size in Figure 2 is too small and the figures are not presented as vector images.

3. The proposed KFusion method is not compared with recent text-to-video retrieval methods from 2024 and 2025. Meanwhile the reported improvements are somewhat limited (most gains less than 1%) compared to previous literature. In page 8, line 426, there is a claim that KFusion uses 55.2 GB and 56.8 GB higher memory than CLIP4Clip in training and inference, and claims 23.2 GB lower memory than CLIP4Clip in inference. These statements are mutually inconsistent.

4. Although the mamba and KAN architectures may offer some efficiency improvements, there are no quantitative comparisons provided to substantiate these claims.

**Questions:**

Please see weaknesses above.

---

### Official Review · Reviewer_rydN · 2025-10-30

**Soundness:** 3
**Presentation:** 2
**Contribution:** 2
**Rating:** 2
**Confidence:** 5

**Summary:**

The paper tackles the task of text-video retrieval. It introduces KFusion to fuse text and video frames with lower computational cost, achieved by employing the KolmogorovArnold-Network-based Bridge module and Text-Frame Mamba. Finally the authors test the proposed approach on several benchmarks.

**Strengths:**

The paper tackles an important problem. The method seems interesting but the I find the sota comparison very weak.

**Weaknesses:**

The comparison with state of the art is very weak and even in this case the results seem marginal. The paper only compares with 2 year old methods and does not include a clear sota in the table. Without a proper comparison, it is very hard to assess the benefits of the current method both in terms of performance and computational efficiency.

Missing citations:
* Jiamian Wang, Guohao Sun, Pichao Wang, Dongfang Liu, Sohail Dianat, Majid Rabbani, Raghuveer Rao, Zhiqiang Tao; Proceedings of the IEEE/CVF Conference on Computer Vision and Pattern Recognition (CVPR), 2024, pp. 16551-16560
+ most of the papers referenced in the tables of the above method.

**Questions:**

What are the computational savings comparing to other methods? line 426 provides some hints, but are there are also any timing benefits? XPool is a relatively old method, are there any memory benefits comparing to more recent methods?

---

### Note · Authors · 2026-01-15

I have read and agree with the venue's withdrawal policy on behalf of myself and my co-authors.